# The Functional Diversity of the High-Affinity Nitrate Transporter Gene Family in Hexaploid Wheat: Insights from Distinct Expression Profiles

**DOI:** 10.3390/ijms25010509

**Published:** 2023-12-29

**Authors:** Petros P. Sigalas, Peter Buchner, Alex Kröper, Malcolm J. Hawkesford

**Affiliations:** 1Rothamsted Research, West Common, Harpenden AL5 2JQ, UK; buchnerp58@gmail.com (P.B.); malcolm.hawkesford@rothamsted.ac.uk (M.J.H.); 2Faculty of Agronomy, University of Hohenheim, 70599 Stuttgart, Germany; alex.kroeper@uni-hohenheim.de

**Keywords:** NRT2, NRT3, nitrate, nitrogen, wheat, gene expression, phylogeny, regulation

## Abstract

High-affinity nitrate transporters (NRT) are key components for nitrogen (N) acquisition and distribution within plants. However, insights on these transporters in wheat are scarce. This study presents a comprehensive analysis of the NRT2 and NRT3 gene families, where the aim is to shed light on their functionality and to evaluate their responses to N availability. A total of 53 *NRT2s* and 11 *NRT3s* were identified in the bread wheat genome, and these were grouped into different clades and homoeologous subgroups. The transcriptional dynamics of the identified *NRT2* and *NRT3* genes, in response to N starvation and nitrate resupply, were examined by RT-qPCR in the roots and shoots of hydroponically grown wheat plants through a time course experiment. Additionally, the spatial expression patterns of these genes were explored within the plant. The *NRT2s* of clade 1, *TaNRT2.1-2.6*, showed a root-specific expression and significant upregulation in response to N starvation, thus emphasizing a role in N acquisition. However, most of the clade 2 *NRT2s* displayed reduced expression under N-starved conditions. Nitrate resupply after N starvation revealed rapid responsiveness in *TaNRT2.1-2.6*, while clade 2 genes exhibited gradual induction, primarily in the roots. *TaNRT2.18* was highly expressed in above-ground tissues and exhibited distinct nitrate-related response patterns for roots and shoots. The *TaNRT3* gene expression closely paralleled the profiles of *TaNRT2.1-2.6* in response to nitrate induction. These findings enhance the understanding of NRT2 and NRT3 involvement in nitrogen uptake and utilization, and they could have practical implications for improving nitrogen use efficiency. The study also recommends a standardized nomenclature for wheat *NRT2* genes, thereby addressing prior naming inconsistencies.

## 1. Introduction

Nitrogen (N) is an essential macronutrient for plant growth and development as it is a fundamental component of nucleic acids, amino acids, chlorophyll, and cell structural components. As a result, N nutrition has a substantial impact on plant metabolism, growth, and productivity. Despite being abundant in the atmosphere, N is often a major limiting factor for crop productivity due to its low availability in soil. Therefore, plants have evolved sophisticated mechanisms to sense, acquire, and utilize the available soil N, including the uptake of inorganic N from the soil through the roots and the distribution within the plant through specialized transporters [1].

N is predominately taken up from the soil as inorganic N in the form of ammonium or nitrate. Although ammonium is the preferred form of N over nitrate for many plants species, as it requires less energy to be assimilated than nitrate, the latter is the predominant form of N taken up by most crop plants because of its higher availability in agricultural soils [2]. The uptake of nitrate is facilitated primarily by nitrate transporters (NRT) located in the plasma membrane of the root cells. Among the members of the NRT family, two subfamilies contribute the most in N uptake, namely nitrate transporter 1/peptide family (NPF) and nitrate transporter 2 (NRT2). The members of the NPF family are involved in the low-affinity transport system (LATS), operating at high external nitrate concentrations (>1 mM) [3]. The NPF family is a large family consisting of 53 genes in Arabidopsis (*Arabidopsis thaliana*), and recent work on wheat (*Triticum aestivum*) has shown that there are 331 genes encoding the putative NPF present in the wheat genome that has been grouped into eight different subfamilies [4,5,6]. On the other hand, NRT2 are high-affinity nitrate transporters responsible for the uptake of nitrate under low external nitrate concentrations (<1 mM). An exception is the NPF6.3 in Arabidopsis, which has been shown to be involved in both low- and high-affinity transport (HATS) systems [7]. Apart from nitrate uptake from the soil, NRT transporters have been shown to be involved in the internal translocation of N, or to act as N sensors, while some members of the NPF family are also involved in the transport of proteins, hormones, and other compounds [1].

Due to its importance in nitrate uptake, the NRT2 family has been extensively studied in different plant species. To date, seven *NRT2* genes have been identified in Arabidopsis and Brachypodium (*Brachypodium distachyon*) genomes [8,9], while only four *NRT2* genes have been found in rice (*Oryza sativa*) [10]. NRT2 transporters are 500–600 amino acids long and have a characteristic conserved core structure of 12 transmembrane domains. Members of the NRT2 family transport nitrate into the cytoplasm of the root cells against the concentration gradient via the proton motive force [11]. Most of the *NRT2* genes have been found to be expressed exclusively in roots, for example *AtNRT2.1* and *AtNRT2.2* in Arabidopsis, thus suggesting their involvement in the nitrate acquisition from the soil. Studies have shown that AtNRT2.1 and AtNRT2.2 account for up to 80% of nitrate influx in Arabidopsis [12]. However, *AtNRT2.4* and *AtNRT2.5* are expressed additionally in aerial parts, while *AtNRT2.7* is leaf-specific in Arabidopsis, thereby suggesting that NRT2s are involved in the N distribution within the plant [13,14,15]. The expression of *NRT2s* is highly responsive to both internal and external cues. In fact, the expression of most of the *NRT2s* has been found to rapidly respond to changes in the external concentration of nitrate as part of a primary response [16,17]. Okamoto et al. (2003) introduced a classification of the Arabidopsis NRT2 into three types based on their response to nitrate provision: nitrate-inducible, -repressive, or -constitutive [8]. In addition to external nitrate concentration, other signals controlling *NRT2* expression include plant nitrogen status/demand controlled by feedback regulation, while other studies have demonstrated a relationship between NRT2 and carbon status or photosynthesis [18,19].

As has been shown in many species, NRT2 transporters require the co-expression of members of another NRT gene family, namely NRT3 (also known as nitrate assimilation related 2 (NAR2)), to function properly [20]. *NRT3* genes encode a small protein of about 200 amino acids that, unlike other NRTs, do not transport nitrate across the plasma membrane. Instead, NRT3 interact with other members of the NRT family, such as the NRT2, modulating their activity and forming what is known as a two-component transport system [21]. In fact, the *Atnrt3.1* mutant showed significantly impaired HATS, indicating that NRT3.1 is required for the normal function of HATS; however, this was not the case for the LATS, which were shown to be independent of NRT3.1 [20].

Although considerable progress has been made in understanding the function and regulation of nitrate transporters in plant species such as in Arabidopsis and rice, there are still many unanswered questions regarding their role and regulation in crop species such as bread (hexaploid) wheat. Wheat is one of the most important crops in temperate latitudes that demands high amounts of N for optimum grain yield and quality. To meet those requirements large amounts of synthetic N fertilizers need to be applied at different growth stages [22]. However, only a fraction of the applied N is recovered by plants, thus causing environmental and economic issues. As a result, more sustainable practices are required to mitigate these issues. One strategy to reduce N inputs could be the development of varieties with improved nitrogen use efficiency (NUE) [23,24]. NUE is the result of the combination of N uptake efficiency and N utilization efficiency. However, those processes are highly complex and tightly regulated; hence, a comprehensive knowledge of all the systems involved in N uptake, transport, and metabolism is required. Previously, the NPF low-affinity transporters were described in wheat [4,25]. However, less has been reported for the members of NRT2 and NRT3 families. Hence, the aim of this study, is to identify and characterize the phylogenetic relations of the hexaploid wheat *NRT2* and *NRT3* genes compared with their orthologs from Arabidopsis, Brachypodium, rice, and barley (*Hordeum vulgare*). Three recent studies have also focused on the *NRT2s* in wheat [26,27,28]; however, each study suggested different nomenclature for the identified *NRT2s*. The presence of different names for the same gene accession can cause confusion and is a major problem in the wheat research community. In this paper, after identifying the *NRT2* genes present in the wheat genome, we identified the different homoeologous subgroups and we suggested a new nomenclature, thereby aligning these with the different names from previous studies and thus aiming to solve the inconsistency in gene names. In addition, we performed a time course analysis of the transcriptional responses of *NRT2* and *NRT3* genes for N starvation and nitrate induction in the roots and shoots of wheat plants, as well as their expression in different tissues. Our results provided insights into the potential roles of the identified NRT2s in the mechanisms underlying N uptake and use in wheat, i.e., information that will have application in improving NUE.

## 2. Results

### 2.1. Wheat NRT2 and NRT3 Gene Family Genome and Phylogeny Analysis

The IWGSC hexaploid wheat cv Chinese spring genome sequence contains 53 NRT2 family genes. More specifically, 16 *NRT2* were found in the A sub-genome, 17 *NRT2* in the B sub-genome, while the D sub-genome contains 20 *NRT2* genes. The majority of the *NRT2* genes are located on the short arm of chromosome 6 in all three sub-genomes, including genes on chromosome 1, 2, 3, and 7. The different number of *NRT2* genes per sub-genome indicates that not all *NRT2* genes are represented by homoeologous genes in all three sub-genomes.

The *NRT2* genes on the short arm of chromosome 6 consist of three clusters. A cluster of the neighboring *TaNRT2.1-2.6*, a second cluster by *TaNRT2.7-2.9*, and a third cluster of *TaNRT2.10-2.14B*. *TaNRT2.9* is missing on chromosome 6A, and *TaNRT2.12* and *TaNRT2.14C* are absent from chromosome 6A and 6B. The *TaNRT2.1-2.6* cluster genomic DNA is inverted on chromosome 6A in comparison with the clusters on chromosomes 6B and 6D (Figure 1, Table 1). *TaNRT2.15* is located on chromosome 2A and 2D but is missing on chromosome 2B. Interestingly a *TaNRT2.15* ortholog is present in both the A and B sub-genomes of the tetraploid wild emmer (TRIDC2AG008290, TRIDC2BG010000). In the wheat cv Chinese spring genome, two *TaNRT2.17* genes were identified, one on chromosome 1D and one with a low confidence unknown chromosome localization. The genome sequencing of wheat cv Stanley (TraesSTA1B03G00193270) and cv Arinalrfor (TraesARI1B03G00196760) verified the localization on chromosome 1B with no homoeologous genes in the A sub-genome. The phylogenetic analysis of the coding DNA sequences separates the wheat NRT2 genes into three distinct clades (Figure 2, Appendix A). Clades 1 and 2 represent the gene clusters found on chromosome 6, with clade 1 consisting of *TaNRT2.1-2.6* and clade 2 of *TaNRT2.7-2.14C*, with subclades for *TaNRT2.7-2.9* and *TaNRT2.10-2.14C*. The phylogenetic analysis reflects the homoeologous relationships of the A, B, and D NRT2 genes, apart from *TaNRT2.14A-C* (which had no clear A, B, and D sub-genome homoeologous separation due to their very high sequence identity). The wheat genome database automated gene annotation [29,30] did not recognize all NRT2 sequences present in the genome. *TaNRT2.2-6B* and *TaNRT2.5-2D* were only annotated as low-confidence genes. Careful sequence analysis confirmed the presence of *TaNRT2.2-6D*. *TaNRT2.5-6D* has a 7 base deletion in the coding region leading to a coding frameshift and changed stop codon, thereby resulting in a truncated 207 amino acid and a potentially non-functional 207 amino acid protein. For *TaNRT2.9*, only partial gene sequences have been identified on chromosome 6B and 6D. For both genes, the identified coding sequence was located behind a potential intron based on the potential intron border sequence, but the approximate 311 nucleotide that is missing the 5′- region has not yet been found in the genome sequence. Gene expression analysis by RT-qPCR, including PCR fragment cloning and sequencing, have confirmed *TaNRT2.9* transcript, thus suggesting an actively expressed NRT2 gene.

A total of 11 NRT3-encoding genes were identified in the hexaploid wheat genome, forming four homoeologous subgroups (Table 1). These genes were found to be phylogenetically closely related to the *NRT3* (*NAR2*) genes from barley, while they exhibited greater genetic divergence from those of other species (Appendix A). The *TaNRT3.1* and *TaNRT3.2* genes are located on the long arm of chromosome 6 across all sub-genomes, while the two homoeologs of *TaNRT3.3* were found on chromosome 5B and 5D, with the A sub-genome homoeolog situated on chromosome 4 (Figure 1). Additionally, our sequence analysis revealed the presence of two additional NRT3-encoding genes, *TaNRT3.4*, on chromosome 5 of the B and D sub-genomes; however, these are present only in certain cultivars (such as cv Mace, cv Jagger, cv Stanley, and others) and are absent from the cv Chinese spring genome.

### 2.2. Phylogenetic Relationship of NRT2 from Wheat and Other Plant Species

To explore the phylogenetic relationship of bread wheat *NRT2* genes and their orthologs in various other plant species, a phylogenetic tree was constructed using the protein coding DNA sequences and the 200 bp 3′ noncoding region of the *NRT2* genes found in wheat D sub-genome, thereby representing all the different wheat homoeologous subgroups. Also included are the sequences from the well-studied species Arabidopsis, Brachypodium, rice, and barley (refer to Appendix A). This phylogenetic analysis categorized the NRT2 genes into distinct subfamilies.

Based on the phylogenetic tree (Figure 3, Appendix A), the *NRT2* genes may be divided into subfamilies. Subfamily 1 only includes members from wheat (*TaNRT2.1-2.6*), Brachypodium, and barley. Similarly, Subfamily 2 consists of 10 *NRT2* wheat homologs and sequences from Brachypodium and barley. All the Arabidopsis and rice *NRT2* genes fall into Subfamily 3, along with four sequences from wheat (*TaNRT2.15-2.18*). The phylogenetic distances in Subfamily 3 were larger in comparison to the ones in Subfamilies 1 and 2. Subfamilies 1 and 2 were found to be phylogenetically close to *OsNRT2.1* and *OsNRT2.2*. More separated was *TaNRT2.18*, which is closely related to *OsNRT2.4* and *AtNRT2.7*. *TaNRT2.16* and *2.17* fall into a separate branch of Subfamily 3 along with *BdNRT2.5*, *HvNRT2.1*, and *OsNRT2.3*. Finally, *AtNRT2.1-2.4* and *AtNRT2.6* formed an isolated cluster within Subfamily 3.

### 2.3. Transcriptional Regulation of NRT2 and NRT3 in Response to N Starvation

To elucidate the distinct functional roles of the identified *NRT2* genes in wheat, a time course analysis of expression profiles in response to N starvation was conducted for root and shoot tissues. The gene expression analysis by RT-qPCR was performed on hydroponically cultivated plants at three time points, i.e., 1, 3, and 6 days, after the initiation of N starvation.

Analysis of the root tissue revealed distinctive patterns of gene expression within the NRT2 gene family (Figure 4, Appendix A). Within clade 1 of the NRT2 gene family, comprising *TaNRT2.1-2.6*, a notable upregulation in gene expression was observed in response to N starvation within 1 day. Specifically, the expression of *TaNRT2.1* exhibited more than a 32-fold increase, stabilizing at elevated levels compared to plants under high-N conditions. Similarly, the mRNA levels of *TaNRT2.2* were found to be significantly higher after prolonged N starvation compared to control plants. However, the other members within clade 1 (*TaNRT2.3-2.6*), while exhibiting an initial induction in expression 1 day after N starvation, showed fluctuations in mRNA accumulation over time. Notably, *TaNRT2.1* was the most highly expressed among the NRT2 clade 1 genes during N starvation, with levels that were 3- and 7-times higher than those of *TaNRT2.2* at 1 and 3 days, respectively, when following the onset of N starvation.

A distinct response pattern emerged among the majority of the NRT2 clade 2 genes in the roots. In contrast to the genes of clade 1, *TaNRT2.10-2.14* exhibited a marked reduction in mRNA accumulation within the first day of N starvation, and their expression levels remained consistently lower than those of plants subjected to high-N conditions across all examined time points. Additionally, the expression of *TaNRT2.10-2.14* showed an increase over time in the high-N-treated plants, which is more likely to be developmentally related. However, the mRNA levels in the N-starved plants did not follow this pattern, and they were found to be significantly lower at all the examined time points. *TaNRT2.7*, *2.8*, and *2.9* did not exhibit any pronounced response to N limitation in the root tissues.

The *NRT2s* of clade 3 had varied responses to N starvation. *TaNRT2.15* exhibited a gradual suppression in roots, with expression levels reaching the lowest level, i.e., 32-fold lower, at 6 days after N starvation. Lastly, *TaNRT2.16* and *2.18* demonstrated a sharp induction soon after exposure to N starvation, with their expression remaining at significantly higher levels compared to the high-N-treated plants.

For shoots, the transcriptional responses were less pronounced when compared to the responses observed for root tissues (Figure 5, Appendix A). Specifically, clade 1 genes showed an induction in expression levels within 1 day of N starvation, followed by a subsequent decline that brought their expression back to levels comparable to those observed in plants under high N conditions. The only exception was *TaNRT2.3*, which showed no significant difference in expression between the two N treatments. As was also observed for the roots, *TaNRT2.1* displayed the most substantial increase, showing a 4-fold induction within 1 day of N starvation. Notably, the observed transcript abundance of *TaNRT2.1-2.6* in the shoots was lower, indicating their likely greater functional role in root nitrate uptake.

In contrast, the remaining *NRT2* genes exhibited no significant response to N starvation in the shoots, with expression levels showing no significant difference between the two N treatments at the examined time points following N starvation. Only *TaNRT2.18* displayed a response similar to that observed in the roots, showing a 12-fold increase within 1 day of N starvation. The expression of *TaNRT2.18* remained at significantly higher levels than that found in the shoots of high-N-treated plants. It is noteworthy that the *TaNRT2.18* transcript abundance was higher in the shoots compared to the roots, a trend that was also observed only in the case of *TaNRT2.8*.

Within the NRT3 family, *TaNRT3.3* exhibited an increase in expression soon after exposure to N starvation, followed by a subsequent decline, thereby gradually reaching levels similar to those in the high-N-treated plants.

### 2.4. Transcriptional Regulation of NRT2 and NRT3 to Nitrate Provision/Induction

The expression levels of *NRT2s* were monitored in response to the nitrate resupply to N-starved plants. The analysis was performed at 3, 6, 12, 24, and 72 h after nitrate provision for both the root and shoot tissues (Figure 6 and Figure 7, Appendix A). The expression data from the plants grown under constant high N supply were also analyzed as a reference for selected time points.

In the roots, the *NRT2s* of clade 1 shared the same expression pattern in response to nitrate resupply. More specifically, *TaNRT2.1-2.6* exhibited a substantial induction 3 h after nitrate resupply, with the increase reaching as high as 130-fold in the cases of *TaNRT2.3* and *2.4*. However, expression decreased substantially in the subsequent 24 h, and this continued to gradually further decrease until 72 h after nitrate resupply.

The expression profiles of *TaNRT2.7-2.9* displayed fluctuations over time without displaying any significant deviations when compared to the plants treated with high N. *TaNRT2.11* and *2.13* shared a similar expression profile in the root tissues, with an initial decrease recorded within the first 6 h after N-starved plants were exposed to nitrate. Subsequently, their mRNA levels gradually increased, eventually reaching similar levels to those observed in high-N-treated plants. The short-term decrease was only observed for *TaNRT2.11* and *2.13*, thereby indicating a specific regulatory mechanism. *TaNRT2.10*, *2.12*, and *2.14* demonstrated a steady increase in mRNA accumulation over time.

*TaNRT2.15* exhibited a distinct expression profile, one that was not observed for any other NRT2 gene. Initially, expression levels were more than 16-fold lower in N-starved plants. However, the expression increased significantly within 3 h of nitrate provision, which was subsequently maintained at levels similar to those recorded in the roots of high-N-treated plants. *TaNRT2.16-2.18* expression displayed a progressive decline, stabilizing within 24 h after nitrate resupply.

Finally, the transcriptional response of the NRT3 family members closely paralleled those observed for *TaNRT2.1-2.6*. Specifically, a rapid surge in expression was detected for all three *NRT3* genes within 3 h following nitrate supply, which was followed by a more gradual decline over time.

In the shoots, *TaNRT2.1-2.6* exhibited a significant response to nitrate resupply similar to the expression profiles observed for roots. All genes of clade 1 displayed an induction of expression within the first 3 h following nitrate resupply, which was followed by a gradual decline in transcript abundance. This decline after the initial increase was more pronounced up to 12 h after nitrate resupply.

While some of the responses were noted in the expression levels of the NRT2 clade 2 genes upon exposure to nitrate, only *TaNRT2.8* demonstrated significant changes over time. *TaNRT2.16* and *2.18* were induced within the initial 3 h following nitrate resupply, which was followed by a decline. Notably, this pattern was different from what was observed in the roots, where the expression did not exhibit any spike after nitrate provision.

The expression of the *NRT3* genes increased after nitrate resupply and remained at elevated levels until 6 h post-N provision, after which it began to decline gradually. Eventually, it reached levels similar to those in plants subjected to a constant high-N supply within 24 h.

### 2.5. Spatial Expression Analysis of the NRT2 and NRT3 in Field-Grown Wheat

Examining the spatial expression patterns of the identified *NRT2* and *NRT3* genes is important for understanding the underlying mechanism of nitrate uptake and distribution in wheat. In this study, the expression patterns of *NRT2* and *NRT3* were determined in the different tissues of field-grown wheat at anthesis, including the root, stem, flag leaf sheath, flag leaf, rachis, and spike (Figure 8A, Appendix A). Additionally, the expression in the developing grains was determined at three time points during grain filling (Figure 8B).

At anthesis, the *NRT2s* displayed a distinct expression pattern. The genes of NRT2 clade 1 were predominately expressed in roots, with *TaNRT2.1*, *2.3*, and *2.4* having expression exclusively in the roots. The expression of *TaNRT2.2*, *2.5*, and *2.6* was also detected in the above-ground tissues; however, the detected transcript abundances were remarkably lower compared to the root. The abundances of some gene transcripts of clade 1 genes in roots were the highest detected among all of the examined *NRT2* genes in all the different tissues, implying the primary role of the NRT2 clade 1 genes in nitrate uptake from the soil.

In contrast, the NRT2 clade 2 genes were not exclusively expressed in the root, with many of the genes having high expression in the above-ground tissues. Notably, *TaNRT2.7* and *2.8* showed comparable transcript levels in the root, flag leaf, and rachis, suggesting a role in N allocation in the above-ground tissues. *TaNRT2.7* showed the highest transcript abundance in the spike and the rachis when compared to other *NRT2s*, and it was also highly expressed in the stem and the flag leaf, indicating a potential contribution to N allocation in all tissues, including grain nitrogen accumulation. Although *TaNRT2.13* and *2.14B* exhibited the highest expression levels in the roots, they were also amongst the most highly expressed *NRT2* genes in all of the examined above-ground tissues. *TaNRT2.14B* was the most highly expressed in the main stem and also showed a high transcript number in the flag leaf and flag leaf sheath.

For the clade 3 genes, *TaNRT2.18* was the only gene that showed higher expression in the above-ground tissues compared to the roots. More specifically, *TaNRT2.18* was strongly expressed in the flag leaf blade and flag leaf sheath, suggesting a role in the nitrate transport in these tissues. A relatively high expression of *TaNRT2.16* was detected in the roots, as well as in the flag leaf sheath and the rachis.

The genes of the NRT3 family showed expression in all of the examined tissues, with *TaNRT3.3* showing the lowest expression across all of the tissues when compared to *TaNRT3.1* and *3.2*. *TaNRT3.1* was predominately expressed in the roots and was the most highly expressed compared to the other *NRT3* genes in the roots, thereby highlighting an involvement in the nitrate uptake in the roots. In contrast, in the flag leaf sheath, rachis, and spike, *TaNRT3.2* showed the highest transcript abundance.

Only a subset of *NRT2* genes was found to be expressed during grain development, including the genes of clade 2 (*TaNRT2.7* and *2.8*) and clade 3 (*TaNRT2.15*, *2.16*, and *2.18*). No gene of the NRT2 clade 1 showed expression in the grain, suggesting a distinct regulation of nitrate accumulation in the grain. The *NRT2s* expressed in the grain were also found to be expressed in the other above-ground tissues, as described above, thus further supporting their involvement in the nitrate transport around the plant. Lastly, among the NRT3 family, only *TaNRT3.3* exhibited expression in the developing seeds.

## 3. Discussion

### 3.1. The NRT2 Family in Wheat Consist of Multiple Genes Organized in Three Clades

The NRT2 family is a diverse group of high-affinity nitrate transporters, and they play important roles in the N acquisition and utilization in plants. Consequently, elucidating their phylogenetic relationships and functional roles will have implications for understanding the mechanisms governing wheat N utilization efficiency, as well as for their potential applications in improving NUE.

The NRT2 family has been shown to consist of multiple genes in plants; for example, 7 in both Arabidopsis [13] and Brachypodium [9], 4 in rice [10], and 10 in barley [31]. The diversity of *NRT2* genes highlights the complexity of nitrate transport in plants. Recent publications focusing on bread wheat confirmed the presence of multiple genes in the wheat genome [26,27,28]. However, these studies have reported different numbers while proposing distinct nomenclatures, thus leading to inconsistencies and potential confusion in the wheat research community.

In the current study, the *NRT2* and *NRT3* genes were identified in the hexaploid wheat genome, including both high-confidence and low-confidence genes. Furthermore, the homoeologous relationships between the genes present in the A, B, and D wheat sub-genomes have been described. A new nomenclature is proposed that accounts for these homoeologous subgroups (Table 1), and a conversion table is presented, thereby aligning the present nomenclature with that of recent publications [26,27,28]. This resource aims to facilitate future research and avoid inconsistencies in gene names.

Based on the current analysis, the wheat NRT2 family comprises 53 members distributed across the three sub-genomes (A, B, and D), and these were organized into 20 homoeologous subgroups. Wheat showed the highest number of *NRT2* genes compared to the other species (even after consideration of the 3 sub-genome structure), which highlights the complexity of the NRT2 family in wheat and emphasizes the importance of further understanding their functional roles. Wheat *NRT2* genes are not evenly distributed across wheat sub-genomes as only 15 out of the 20 identified homologs are present in all three sub-genomes, while some members are only found on the D sub-genome. The uneven distribution of *NRT2* genes is also apparent among the chromosomes, as over 80% of the identified wheat *NRT2* genes are located on the short arm of chromosome 6, which spans a region of less than 0.8 Mb on chromosome 6A, as well as 1.4 Mb on chromosome 6B and 6D. These genes are further clustered into three subclusters, as shown in Figure 1. Tandem sequences exhibit high sequence homology, which is suggestive of their origin through recent multiple duplication events (Appendix A). These findings are consistent with previous research, suggesting that *NRT2* genes have undergone tandem duplication in wheat and in other plant species [32].

Based on the phylogenetic analysis, the wheat *NRT2* genes clearly separated into three clades (Figure 2). Notably, the clade separation correlated with the physical location of the *NRT2s* across chromosomes, as the genes present on chromosome 6 appeared to form two separate clades, clade 1 and 2, while clade 3 comprised the remaining *NRT2* genes that are located on different chromosomes. The phylogenetic relationship of wheat *NRT2s* with the *NRT2s* from other plant species revealed that the members of clades 1 and 2 fall into the same monocotyledonous-specific branch within the sequences of Brahypodium and barley (Figure 3). In contrast, the members of clade 3 were found to be phylogenetically closely related to the other orthologs from monocotyledonous and dicotyledonous species, thus suggesting a more conserved role across plants.

### 3.2. Distinct Expression Profiles Revealed Specific Functional Roles

Along with the analysis of the phylogenetic relationships of the identified wheat *NRT2* genes, an expression analysis under changing N conditions and a spatial analysis were conducted to elucidate the functional roles of wheat NRT2s. Based on the overall gene expression patterns of the identified *NRT2* genes in hexaploid wheat, it is evident that the genes within the same clade typically exhibit distinct expression profiles when compared to genes from other NRT2 clades. These observations suggest that genes in the same clade may share similar regulatory mechanisms, implying that they may also have similar functionalities. In fact, even the members of the same subcluster of tandem genes present on chromosome 6 showed similar responses to changing N conditions, suggesting that these duplicated sequences share similar transcriptional regulatory mechanisms.

#### 3.2.1. NRT2 Clade 1

The presence of a similar response to changing N conditions was more apparent in the case of NRT2 clade 1 genes. Their expression was induced soon after N starvation in the roots and shoots, which was then gradually suppressed in most cases. An exception was *TaNRT2.1*, which showed significantly higher expression during prolonged N starvation. *TaNRT2.1* was the most highly expressed gene of the NRT2 family in the roots, indicating it has a central role in nitrate acquisition. In addition, the *TaNRT2.1-2.6* genes showed typical responses to N provision, reaching a peak 3 h after the exposure of N-starved plants to nitrate, which was followed by a decline back to pre-exposure levels. This transcriptional response is indicative of genes involved in the primary nitrate response (PNR). PNR is triggered soon after the exposure of plants to nitrate and is essential for plant adaptation to changing N conditions [17]. In Arabidopsis, *AtNRT2.1* has been found to be a target of various transcription factors such as in the NLP proteins LBD37/38/39, TGA1/4, and NIGT1, which act as master regulators of PNR, thereby controlling the transcription of thousands of genes upon exposure to nitrate, including genes involved in nitrate uptake and N assimilation [33,34,35,36]. In agreement, the expression profiles of *TaNRT2.1-2.6* did not seem to correlate with the changes in internal N status (Appendix A), further supporting the idea that their expression is regulated upon sensing the external concentration of N rather than by an internal N status signal. According to the suggested classification of Okamoto et al. (2003), *TaNRT2.1-2.6* genes are categorized as nitrate-inducible genes participating in HATS [8].

The findings presented are in accordance with previous studies reporting the strong induction of *TaNRT2.1* by nitrate [37]. *BdNRT2.1* and *2.2*, the orthologs of the wheat NRT2 genes of clade 1, are predominantly expressed in roots, and their expression is induced by nitrate [9]. In the expression analysis of field-grown wheat, most clade 1 genes exhibited specific expressions in the roots, whilst, in hydroponically grown wheat, the expression in the roots was significantly higher than in the above-ground tissues. Taken together, it may be hypothesized that *TaNRT2.1-2.6* genes play a central role in the NRT2 family in terms of being involved in the direct nitrate acquisition from soil, contributing to HATS. Kumar et al. (2023) demonstrated that the expression of *TaNRT2.1-6B* (reported as *TaNRT2.1-B6*) restored the impaired N influx of *Atnrt2.1* [26]. In addition, Wang et al. (2019) found that Brachypodium *nrt2.1* mutants significantly impair HATS, further supporting this hypothesis [9]. The presence of multiple NRT2s in the same clade sharing similar functionalities, especially apparent in monocots, may provide some plasticity and evolutionary advantage. In fact, members within the same clade can compensate for the loss of function of others, as demonstrated in the Brachypodium *nrt2.1* mutants, where a higher expression of *BdNRT2.6* was recorded, thus partly compensating for the reduced nitrate influx [9].

#### 3.2.2. NRT2 Clade 2

The NRT2 clade 2 genes, specifically *TaNRT2.10-2.14B*, showed an increased expression during growth in the root tissues, while their expression remained relatively stable in the aerial parts of the plant. Notably, the expression levels in the roots were significantly lower in N-starved plants. The provision of nitrate led to a gradual increase in transcript abundance, reaching similar expression levels to those observed in high-N-treated plants within 24 to 72 h. Therefore, these observations suggest a regulatory pattern that implies a demand-driven mechanism governing their transcription. Consistent expression patterns in response to nitrogen exposure were also reported for *HvNRT2.7*, *HvNRT2.8*, and *HvNRT2.9*, which is in alignment with the findings of the present study for the corresponding wheat orthologous genes [31].

It is worth highlighting that certain NRT2 genes of clade 2 exhibited expression in the above-ground plant tissues. Collectively, these findings support the hypothesis that clade 2 genes primarily participate in the process of nitrate translocation from root to shoot—the major site of nitrogen assimilation and tissue with the greatest N-requirement. This hypothesis aligns with the observed downregulation of these genes under N-deprived conditions, reflecting the reduced nitrate translocation from root to shoot. In contrast, their expression increases as the internal nitrogen pool increases following nitrate provision.

An exception to this regulation pattern was identified in the cases of *TaNRT2.7*, *2.8*, and *2.9*, which displayed constitutive expression patterns in response to the changes in N availability. These findings indicate diverse regulatory mechanisms for NRT2 clade 2 genes, which contrasts with the more uniform regulation observed in the case of clade 1 genes. Furthermore, it is noteworthy that *BdNRT2.3* and *BdNRT2.4*, which are phylogenetically closely related to clade 2 NRT2 genes, were previously classified as nitrate-constitutive genes by Wang et al. (2018), thus providing additional evidence for a divergence in the regulatory mechanisms governing clade 2 genes within monocotyledons [1].

#### 3.2.3. NRT2 Clade 3

A similar expression profile in response to changing N conditions was also observed for *TaNRT2.16*, *2.17*, and *2.18*. In roots, they exhibited a strong induction in N-starved plants, while their expression gradually decreased after exposure to a high concentration of nitrate. This expression pattern showed a negative correlation with the changes in the total N content within the roots, indicating that their regulation may be tightly controlled by the internal signals of N status. Specific amino acids like glutamine and glutamate, as well as plant hormones such as cytokinins, act as local or systemic signaling molecules of internal N status, in which they regulate many metabolic and developmental processes [38,39]. In agreement with the results presented here, *BdNRT2.5*, the ortholog of *TaNRT2.16* and *2.17*, also displayed induced expression under N starvation conditions, as well as a sharp decrease upon exposure to nitrate [9].

Although *TaNRT2.16*, *2.17*, and *2.18* share similar regulatory responses to changing N status, there are differences that suggest distinct functional roles among these genes. In fact, *TaNRT2.18* was the only member of the NRT2 family that showed higher expression in shoots than in roots. Furthermore, the response to changing N conditions was more pronounced in the shoots than in the roots, indicating that *TaNRT2.18* might play a crucial functional role in above-ground tissues. This divergence in function among the genes of clade 3 is further supported by the phylogenetic analysis, which suggests that *TaNRT2.16* and *2.17* are phylogenetically distant from *TaNRT2.18*, thus implying possible functional diversification.

The phylogenetic analysis showed that *TaNRT2.16* and *2.17* are orthologs of *OsNRT2.3*, *BdNRT2.5*, and *HvNRT2.1*. Similar to the current findings, previous studies have reported the induction of *HvNRT2.1* and *BdNRT2.5* under N starvation conditions [9,31]. In rice, detailed investigations have revealed that *OsNRT2.3* plays a role in translocating the nitrate from roots to shoots, with localization studies indicating expression in primary and lateral root steles, as well as suggesting involvement in the phloem loading of nitrate [40]. Consequently, it may be speculated that *TaNRT2.16* and *2.17* may be involved in the long-distance transport of nitrate from roots to shoots. Spatial expression analysis indicated their expression in other tissues apart from the roots, such as the seeds, thus suggesting a general role in the translocation of nitrates within the plant.

In contrast, *TaNRT2.18* exhibited a distinctive expression profile compared to the other genes of the wheat NRT2 family. Specifically, while expression was detected in both the roots and shoots, it was higher in the shoots of hydroponically grown wheat. Similarly, in field-grown mature plants, high mRNA levels were detected in the flag leaf and sheath (Figure 8A). Furthermore, expression was also detected in the seeds (Figure 8B). This aligns with the results presented by Feng et al. (2011), who reported that the *TaNRT2.18* rice ortholog, *OsNRT2.4*, is predominately expressed in shoots [18]. Additionally, phylogenetically related genes such as *AtNRT2.7* and *BdNRT2.5* have shown high expression levels in above-ground tissues, indicating a conserved involvement of *TaNRT2.18* orthologs in the nitrate transport in shoots [9,13,14]. In terms of response to changing N conditions, *TaNRT2.18* displayed a unique pattern with a gradual induction during N starvation in shoots, in which it maintained high expression levels during prolonged starvation. This observation supports the hypothesis that a higher expression may indicate an increased demand for N in these tissues. Nitrate resupply led to an upregulation within 6 h, which was followed by a substantial downregulation. This concurs with what has previously been reported by Cai et al. (2008), who reported an initial induction in *OsNRT2.4* levels within 4 h of nitrate feeding, which was then followed by a sharp decline [10]. Based on the expression profile, Wei et al. (2018) suggested that *OsNRT2.4* may be involved in the redistribution of nitrate from older to developing tissues [41]. However, Wei et al. (2018) reported a downregulation of *OsNRT2.4* in response to N starvation in rice, which is an opposite pattern compared to that found for *TaNRT2.18*, thus indicating a divergence in the roles between the two [41]. The differential regulation of the orthologs between wheat and rice might indicate possible species-specific functional differences, which could be attributed to distinct N utilization strategies. The characterization of *AtNRT2.7* has shown that it is localized to the vacuole membrane and expressed in reproductive organs, thereby facilitating nitrate accumulation in the seeds [42]. In contrast, Wei et al. (2018) demonstrated that *OsNRT2.4* is localized in the plasma membrane [41]. Despite the reported differences in subcellular localization, both *OsNRT2.4* and *AtNRT2.7* have significant roles in N remobilization and homeostasis. Therefore, it is proposed that *TaNRT2.18* may also play a crucial role in remobilizing the stored nitrate from older to developing tissues, particularly when under stress conditions and during senescence.

#### 3.2.4. NRT3

*NRT3* genes encode small proteins with two transmembrane motifs that do not directly facilitate nitrate transfer across the plasma membrane. Nevertheless, many studies have highlighted their pivotal role in nitrate uptake. NRT3 proteins function as partner proteins interacting with NRT2, thereby forming what is known as a two-component nitrate transport system [21]. Previous investigations in model plants such as Arabidopsis and rice have demonstrated the essential role of NRT3 in HATS. The Arabidopsis knock-out mutant, *nar2.1*, exhibits impaired growth and reduced N uptake, which is particularly pronounced under low-N conditions [11,21,43]. In both the roots and shoots of wheat, the identified wheat *NRT3* homologs showed very similar responses to N starvation and N resupply as the *NRT2s* of clade 1, which is consistent with the suggestion of interaction to control nitrate uptake. Furthermore, it is not unlikely that TaNRT3 interacts with other members of the NRT2 family. Notably, *Atnrt3* mutants have a stronger phenotypic effect compared to *Atnrt2.1/2.2* mutants, suggesting that NRT3 are not only required for nitrate uptake from the soil, but are also involved in the nitrate transport within plants by interacting with other members of NRT2 family [20,21]. A spatial expression analysis of the field-grown wheat revealed *NRT3* expression in the above-ground tissues, whereas the NRT2 clade 1 genes exhibited root-specific expression, thus suggesting that NRT3 interacts not only with TaNRT2.1-2.6, but also potentially with the genes of clades 2 and 3, which—based on the results presented in this study—are likely key players in N translocation and redistribution. In rice, *OsNRT2.3a* gene, which is involved in N translocation, also requires *OsNAR2* to function effectively [18]. As a result, it is suggested that NRT3s interact with various NRT2 family members and have a broader role in N homeostasis, thus making NRT3 proteins important targets for crop improvement. In fact, transgenic rice plants with increased expression of *OsNAR2.1* (NRT3) via its native promoter showed increased nitrate uptake, yield, and NUE [44].

#### 3.2.5. Implications in Improving the NUE

NUE is a complex trait and the result of the combination of N uptake and N utilization efficiencies. Uptake efficiency refers to the plant’s ability to acquire N from soil, while the remobilization of N during grain filling significantly contributes to N utilization efficiency [23]. NRTs play a pivotal role in both processes. Therefore, exploring allelic variation in NRTs within older varieties, as well as in wild and ancestral germplasms, holds promise for identifying novel alleles that could positively impact these processes, thus ultimately enhancing the NUE. Notably, genetic variation in *NRT1.1B* (*OsNPF6.5*) has been associated with differences in the NUE among rice subspecies [45]. Another approach for improving NUE involves genetic engineering by manipulating the expression of NRTs to regulate N uptake or internal N distribution within the plant. In rice, the overexpression of NRTs, such as *OsNRT2.1* or *OsNRT2.3,* positively affect crop productivity and the NUE [46,47]. The tissue-specific expression and the different expression patterns of identified *NRT2s* in response to the nutritional signals presented in this study underscore the diverse roles of wheat NRT2s in N uptake, internal partitioning, and delivery to seeds. This knowledge is valuable for pinpointing additional promising candidate genes to enhance NUE and crop productivity.

## 4. Materials and Methods

### 4.1. Identification and Phylogenetic Analysis of the NRT2 and NRT3 Genes in the Wheat Genome

*NRT2* and *NRT3* coding sequences and genome localizations were identified from already predicted annotated genes from the wheat cv Chinese spring WGSC, via a RefSeq v1.1 gene annotation INSDC Assembly GCA_900519105.1 in EnsemblPlants [29,30], as well as by an EnsemblPlant Blast analysis of the homoeologous and paralogous wheat genes of other wheat varieties and the phylogenetically closely identified orthologous *NRT2* and *NRT3* cereal genes from Brachypodium and rice. The verification of coding sequences and genome exon-intron structure was performed by using the EMBL-EBI sequence analysis tools [48]. The suggested nomenclature for the identified *NRT2* genes was based on their physical order on chromosome 6D for *TaNRT2.1-2.14* followed by *TaNRT2* genes on chromosomes 2, 3, 1, and 7D (Table 1). The numbering of the *NRT3* genes was based on their physical order on chromosomes 6 and 5D.

Coding of the DNA sequence alignments and the phylogenetic tree building were carried out using MUSCLE [49] and PHYML-100 bootstraps [50] within Geneious^®^ 10.2.3 software (https://www.geneious.com (accessed on 21 December 2023)) with default parameters. For the NRT2 and NRT3 gene families, the phylogenetic trees are based on 53 single wheat NRT2 homoeologs with or without the seven Arabidopsis, six Brachypodium, three rice, and six barley NRT2 coding sequences, as well as the twelve single wheat NRT3 homoeologs with two Arabidopsis, two Brachypodium, two rice, two barley, and four sugarcane (*Saccharum officinarum*) coding sequences (Appendix A).

### 4.2. Secondary Transmembrane Structure Analysis of the NRT2 and NRT3 Protein Sequences

The secondary protein and transmembrane structure was analyzed using the UCL Department of Computer Science PSIPREP server protein analysis platform, especially MEMSAT-SVM and PSIPREP [51,52,53].

### 4.3. Plant Material and Hydroponic Culture

For the hydroponic experiments, *T. aestivum* cv Paragon plants were cultured in a custom hydroponic system in a controlled environment chamber. A complete randomized block design with three blocks and three biological replicates per treatment combination was used. Individual plants were grown and were held by foam buds on top of 1 L black pots containing the nutrient solution. The nutrient solution was aerated throughout the experiment by an aeration pump tubing system. The composition of the standard high N nutrient solution, i.e., modified Letcombe [54], was 1.5 mM of Ca(NO_3_)_2_, 5 mM of KNO_3_, 2 mM of NaNO_3_, 1.5 mM of MgSO_4_, 1 mM of KH_2_PO_4_/K_2_HPO_4_ (pH 5.8), 50 μM of FeEDTA, 500 nM of CuCl_2_, 9.2 μM of H_3_BO_3_, 3.6 µM of MnCl_2_, 100 nM of Na_2_MoO_4_, and 770 nM of ZnCl_2_. The nutrient solution was renewed every 3–4 days. The growing conditions were 20/16 °C day and night temperature, respectively, with a 16 h day length. Lighting was provided by fluorescent bulbs at an intensity of 550 µmol m^−2^ s^−1^. The humidity was stable at 65% during the day and 75% during the night.

Seeds were surface sterilized with a 1:40 bleach solution: dH_2_O (*v*/*v*) for 15 min, followed by five washes with sterilized dH_2_O. The seeds were soaked in sterilized dH_2_O overnight at 4 °C in the dark. Subsequently, the seeds were placed in black boxes with wet filter paper to germinate. Seedlings were transferred to the hydroponic culture system 4 days after sowing. Plants were supplied with a half-strength nutrient solution for 3 days and then supplied with the standard high N nutrient solution (10 mM nitrate). Two weeks after germination, the plants were divided into different treatments (day 0). The +N plants continued to receive a standard high N nutrient solution, while the remaining plants were supplied with nutrient solution without any N (−N). For the N-induction treatment, 6-day N-starved plants were resupplied with a high N nutrient solution. Samples were taken at 0, 3, 6, 7, and 9 days (+N); at 1, 3, and 6 (−N) days; and 3 h, 6 h, 12 h, 24 h, 48 h and 72 h after nitrate resupply. The roots were washed three times in dH_2_O, dried on soft paper towels, and frozen immediately in liquid nitrogen (LN2). The whole shoots were harvested, and the shoot base was also washed three times in dH_2_O, dried, and frozen immediately in LN2 and stored at −80 °C. Each biological replicate consisted of pooled tissue material from two plants to ensure enough material for subsequent analysis.

For the spatial expression analysis, *T. aestivum* cv Cadenza was grown in field trials with a total of 200 kg of N ha^−1^ applied in triplicate repetition (part of the Defra-funded Wheat Genetic Improvement Network Diversity 2018 field trial). The experiment was conducted at Bones Close field [51.807087, −0.377540] at Rothamsted Research, Harpenden, UK in 2017/2018. At anthesis, the roots were excavated from the soil with a garden fork and washed several times using dH_2_O. Excess H_2_O was removed using a soft tissue, and the roots were immediately immerged into LN2. Further, the stems, sheaths, flag leaves, rachis, and whole spikes were separated and immediately frozen in LN2. Additionally, the developing grains were sampled 2, 3, and 4 weeks post-anthesis. All the samples were then stored at −80 °C until further use.

### 4.4. Total RNA Extraction and Total N Analysis

The frozen plant material was ground using a SPEX^®^ SamplePrep 6870 freezer mill (SPEX SamplePrep, Metuchen, NJ, USA) and aliquoted. The total RNA isolation was conducted according to Verwoerd et al. (1989) [55] with the modifications described in Buchner and Hawkesford (2014) [25]. The extracted total RNA quality was evaluated by agarose gel electrophoresis (Appendix A). A280/A260 and A260/A230 ratios were used to evaluate the purity of the nucleic acid measured with a Nanodrop 2000 spectrophotometer (Thermo Scientific, Waltham, MA, USA) (Appendix A).

For the total N content analysis, ground freeze-dried root and shoot samples were measured with the Dumas method using a LECO CN628 Combustion Analyzer (LECO Corporation, St. Joseph, MI, USA).

### 4.5. Quantitative Reverse Transcription-PCR (RT-qPCR)

The first-strand cDNA synthesis was performed with 2 µg of total RNA based on the Invitrogen Superscript III standard protocol (Invitrogen, Waltham, MA, USA) and oligo(dT) primers following the manufacturer’s instructions. For the gene expression analysis by RT–qPCR, the forward and reverse primers were designed to allow amplification of all three homoeologs of each targeted *NRT2* and *NRT3* gene (refer to Appendix A). The primer design tool, Primer 3 v2.3.7 (as implemented in within Geneious^®^ 10.2.3 software (https://www.geneious.com (accessed on 21 December 2023))), was used for designing primers close to the 3ʹ end of the coding sequence. Each set of the designed primers (forward and reverse) was tested for specificity and efficiency in the amplification by using a dilution series of template cDNA. For quantitative specific *NRT2/NRT3* gene expression standards, corresponding PCR fragments were amplified using a REDTaq^®^ ReadyMix™ PCR Reaction Mix (Sigma-Aldrich, Dorset, UK), which was cloned into a pGEM-TE plasmid vector (Promega, Madison, WI, USA) and sequenced using the TubeSeq service (Eurofins Genomics, Luxemburg, Luxemburg). *NRT2*/*NRT3* gene-specific plasmid standard dilution series in the region of 0.001 to 25 pgμL^−1^ were prepared and tested beforehand by RT-qPCR.

Real-time PCR was performed in an ABI-7500 real-time PCR system (Applied Biosystems, Waltham, MA, USA) using a SYBR^®^ Green JumpStart™ Taq ReadyMix™ (Sigma-Aldrich, Dorset, UK), as described in Sigalas et al. (2023) [56]. Based on the standard curve Ct values and the known plasmid-specific NRT-PCR fragment molecular weight, the NRT-specific quantitative gene expression was calculated and defined as a single cDNA molecule per µL of cDNA.

### 4.6. Statistical Analysis and Visualization

The mean values and SEs were calculated from three biological replicates. The statistically significant effect of N starvation and nitrate resupply between the different tissues and grain development on the *NRT2* and *NRT3* genes’ expression levels was assessed with analysis of variance (ANOVA) on log2-transformed quantitative RT-qPCR expression data. The N starvation and nitrate resupply effects on the root and shoot N content was also assessed by ANOVA. Statistical analysis was performed in the GenStat statistical software package (21st edition). Following the ANOVA, Fisher’s least significant difference (LSD) at 5% (*p* = 0.05) was calculated based on the SE of the difference between the means of the residual degrees of freedom from the ANOVA, and this was used to compare the relevant group means.

The figures and graphs were created in GraphPad Prism v9.3.1 for Windows. For the generations of heatmap, the package pheatmap v1.0.12 in R Statistical Software v4.1.1 was used.

## 5. Conclusions

In this study, a comprehensive analysis of the wheat NRT2 and NRT3 gene families was conducted, in which their phylogenetic relationships and chromosome locations were explored. Furthermore, a clear nomenclature was presented to account for the homoeologous subgroups. The separation of the *NRT2* genes into three clades was correlated with their physical position across the chromosomes. The expression analysis under varying N conditions revealed shared expression patterns within the same clade in many cases, while distinct expression profiles emerged between the different clades. This observation suggests the presence of diverse regulatory mechanisms and the potential functionalities of the different *NRT2* genes. Most of the *NRT2* genes showed strong expression in root tissue, implying their pivotal role in nitrate uptake. However, a subset of *NRT2* genes showed strong expression in the above-ground tissues, indicating their involvement in nitrate homeostasis and distribution within the plant. This study set the foundation for further research that aims at dissecting the specific functions of individual *NRT2* genes. Additionally, it provides possible targets for crop improvement to optimize N uptake and utilization in wheat through the manipulation of gene expression or the discovery of novel alleles in diverse germplasms.

## Figures and Tables

**Figure 1 ijms-25-00509-f001:**
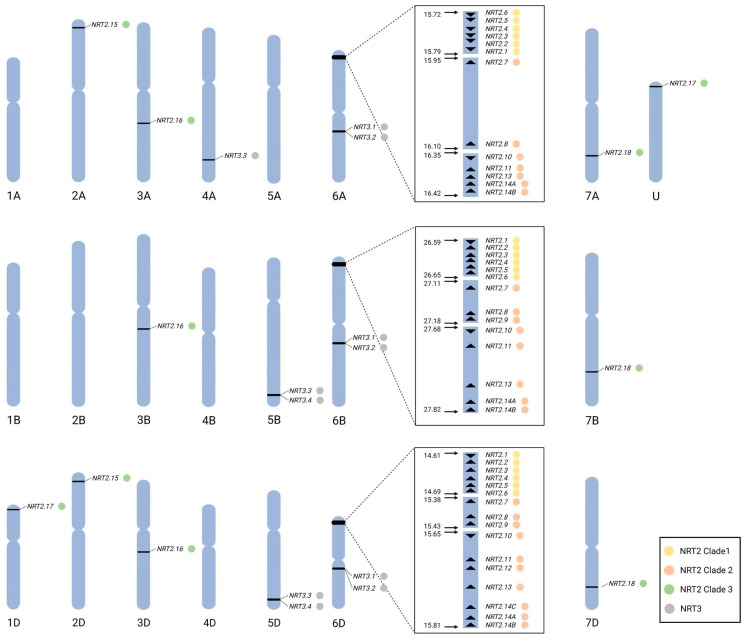
Distribution of the *NRT2* and *NRT3* genes on hexaploid wheat chromosomes. The annotation next to the gene name represents each gene family/clade. The actual genomic location of the *NRT2* and *NRT3* genes can be found in Table 1. Illustration created with BioRender.

**Figure 2 ijms-25-00509-f002:**
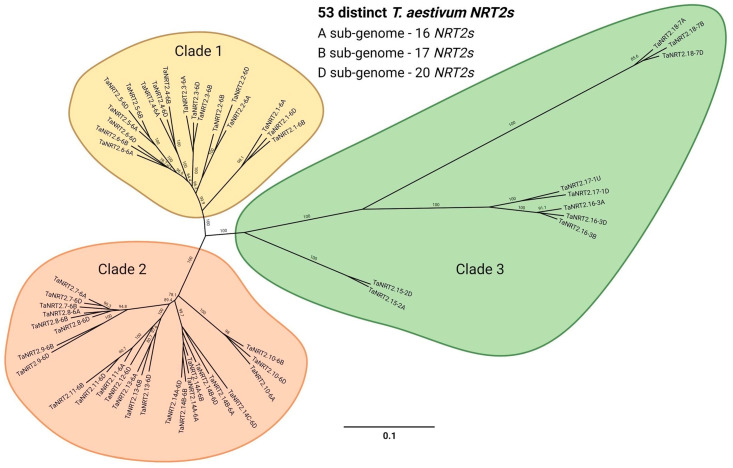
Phylogenetic relationships of the *NRT2* genes of hexaploid wheat. The protein coding DNA sequence plus 200 bp of the 3′-noncoding region of 53 distinct *NRT2* genes were aligned using MUSCLE sequence alignment, and the tree was constructed using the neighbor-joining method. The bootstrap values, expressed as a percentage, were obtained from 1000 replicates. The scale bar corresponds to genetic distance, expressed as the number of nucleotide substitutions per site. The genes were clustered in three clades and are highlighted with different colors. The accession number of the sequences used in the analysis can be found in Table 1, and the percentage identity matrix can be found in Appendix A.

**Figure 3 ijms-25-00509-f003:**
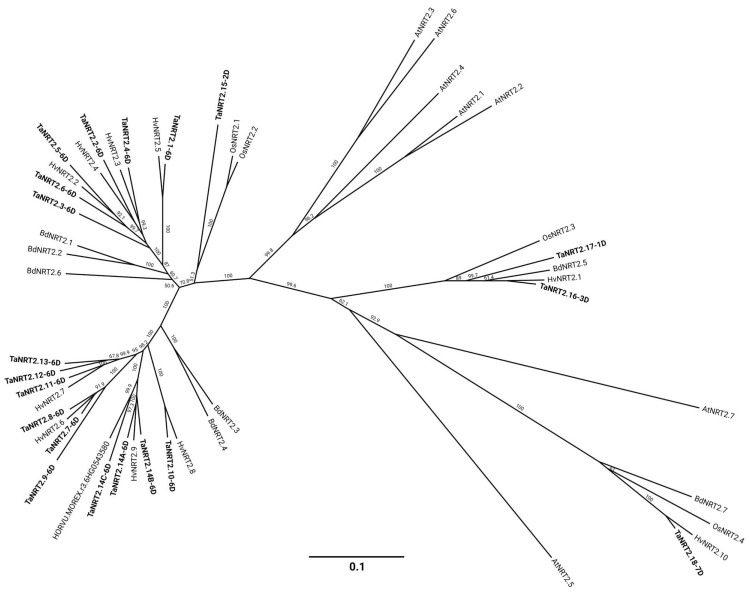
Phylogenetic relationships of the *NRT2* genes of hexaploid wheat, rice, barley, Brachypodium, and Arabidopsis. For simplicity, 20 of the *NRT2* genes present in the wheat D sub-genome were included, thereby representing the identified wheat homoeologous subgroups along with 4 rice, 11 barley, 7 Brachypodium, and 7 Arabidopsis, which were previously identified as *NRT2* genes. The protein coding DNA sequences plus the 200 bp of the 3′-noncoding region were aligned using MUSCLE sequence alignment, and the tree was constructed using the neighbor-joining method. The bootstrap values, expressed as a percentage, were obtained from 1000 replicates. The scale bar corresponds to genetic distance, expressed as the number of nucleotide substitutions per site. The accession number of the sequences used in the analysis can be found in Appendix A, and the percentage identity matrix can be found in Appendix A.

**Figure 4 ijms-25-00509-f004:**
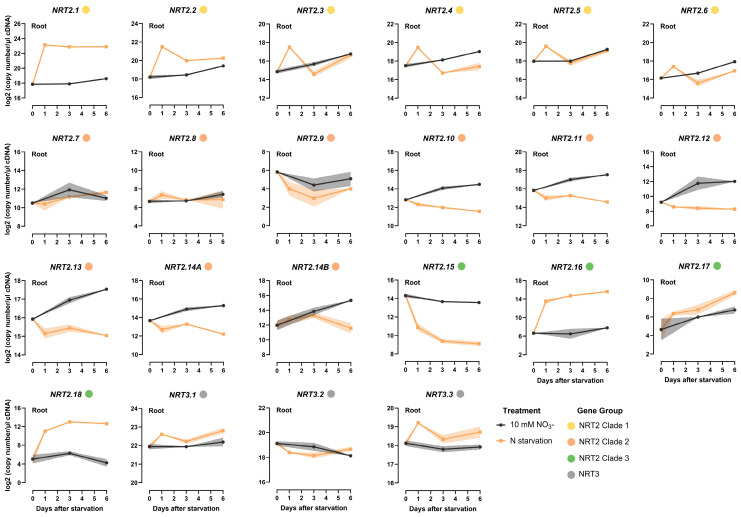
Time course analysis of the *NRT2* and *NRT3* gene expression response to N starvation in the root of wheat (cv Paragon). Plants were introduced to N starvation 2 weeks after germination, and the response was monitored at different time points after starvation (1, 3, and 6 days after). The values are the means of three biological replicates, and the shaded area corresponds to the standard error bands. The Fisher’s LSDs (5%) were as follows: NRT2.1 0.37; NRT2.2 0.39; NRT2.3 0.71; NRT2.4 0.69; NRT2.5 0.51; NRT2.6 0.52; NRT2.7 1.51; NRT2.8 1.24; NRT2.9 2.18; NRT2.10 0.34; NRT2.11 0.28; NRT2.12 1.14; NRT2.13 0.25; NRT2.14A 0.42; NRT2.14B 0.92; NRT2.15 0.70; NRT2.16 1.63; NRT2.17 1.85; NRT2.18 1.39; NRT3.1 0.43; NRT3.2 0.62; NRT3.3 0.61.

**Figure 5 ijms-25-00509-f005:**
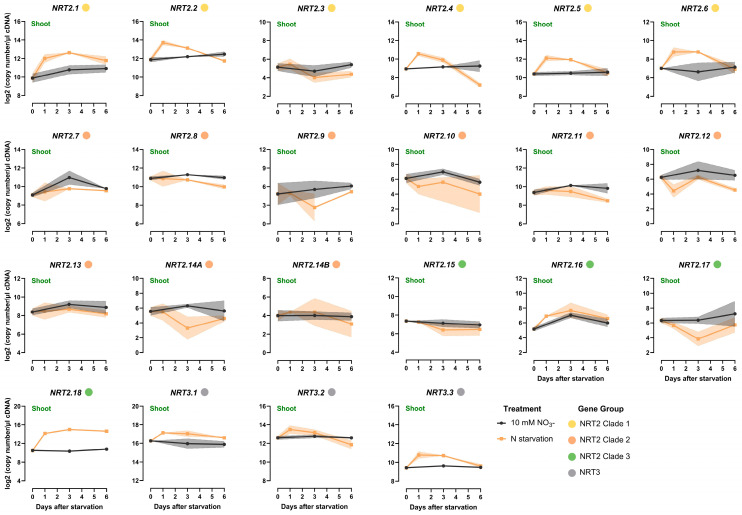
Time course analysis of the *NRT2* and *NRT3* gene expression response to N starvation in the shoot of wheat (cv Paragon). Plants were introduced to N starvation 2 weeks after germination, and the response was monitored at different time points after starvation (1, 3, and 6 days after). The values are the means of three biological replicates, and the shaded area corresponds to the standard error bands. The Fisher’s LSDs (5%) were as follows: NRT2.1 1.32; NRT2.2 0.58; NRT2.3 1.03; NRT2.4 0.98; NRT2.5 0.68; NRT2.6 1.73; NRT2.7 1.73; NRT2.8 1.19; NRT2.9 3.72; NRT2.10 2.60; NRT2.11 1.19; NRT2.12 2.18; NRT2.13 1.33; NRT2.14A 2.61; NRT2.14B 2.98; NRT2.15 1.43; NRT2.16 1.75; NRT2.17 2.91; NRT2.18 0.23; NRT3.1 0.93; NRT3.2 0.99; and NRT3.3 0.60.

**Figure 6 ijms-25-00509-f006:**
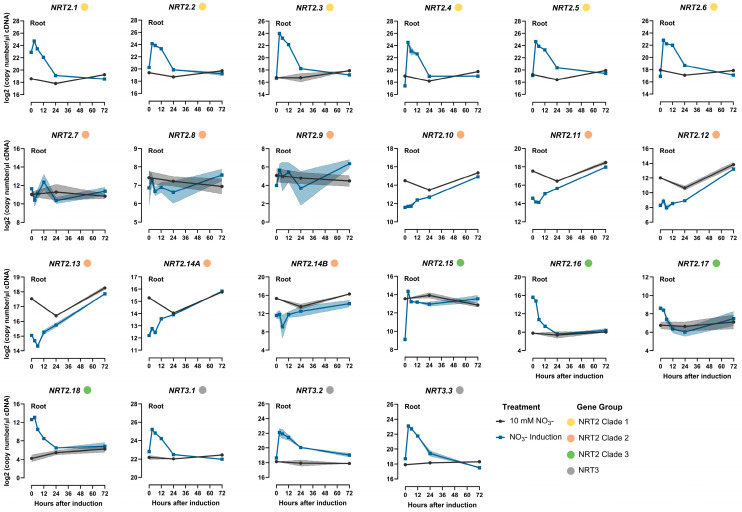
Time course analysis of the *NRT2* and *NRT3* gene expression responses to nitrate induction in the roots of wheat (cv Paragon). Plants starved of N plants for 6 days were supplied with 10 mM of nitrate, and the response was monitored at different time points after nitrate provision (3, 6, 12, 24, and 72 h after supply). For comparison, the data from plants growing in 10 mM nitrate were included. The values are the means of three biological replicates, and the shaded area corresponds to the standard error bands. The Fisher’s LSDs (5%) were as follows: NRT2.1 0.26; NRT2.2 0.41; NRT2.3 0.75; NRT2.4 0.79; NRT2.5 0.30; NRT2.6 0.37; NRT2.7 1.64; NRT2.8 1.38; NRT2.9 3.56; NRT2.10 0.32; NRT2.11 0.19; NRT2.12 0.73; NRT2.13 0.21; NRT2.14A 0.24; NRT2.14B 2.86; NRT2.15 0.63; NRT2.16 0.95; NRT2.17 1.54; NRT2.18 1.44; NRT3.1 0.40; NRT3.2 1.04; and NRT3.3 0.60.

**Figure 7 ijms-25-00509-f007:**
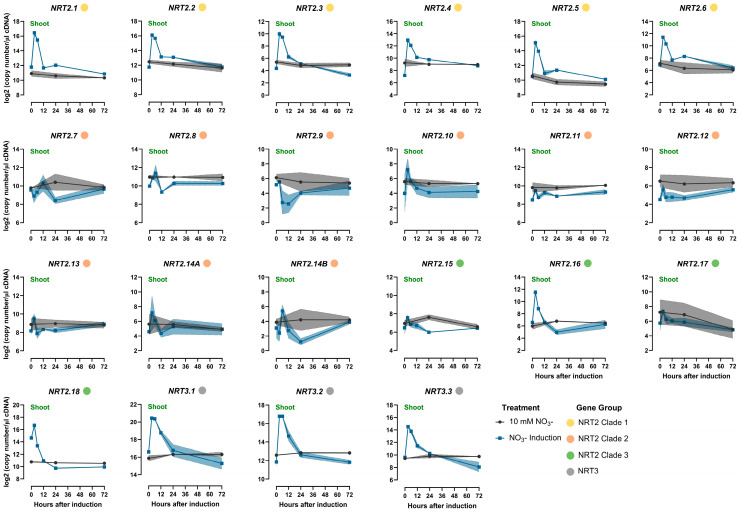
Time course analysis of the *NRT2* and *NRT3* gene expression responses to nitrate induction in the shoots of wheat (cv Paragon). Plants starved of N for 6 days were supplied with 10 mM of nitrate, and the response was monitored at different time points after nitrate provision (3, 6, 12, 24, and 72 h after supply). For comparison, the data from plants growing in 10 mM of nitrate were included. The values are the means of the three biological replicates, and the shaded area corresponds to the standard error bands. The Fisher’s LSDs (5%) were as follows: NRT2.1 0.74; NRT2.2 0.75; NRT2.3 0.95; NRT2.4 0.85; NRT2.5 0.90; NRT2.6 1.31; NRT2.7 1.60; NRT2.8 1.02; NRT2.9 3.18; NRT2.10 3.05; NRT2.11 0.73; NRT2.12 1.71; NRT2.13 1.13; NRT2.14A 3.61; NRT2.14B 2.91; NRT2.15 1.04; NRT2.16 1.27; NRT2.17 3.09; NRT2.18 0.31; NRT3.1 0.97; NRT3.2 0.86; and NRT3.3 1.03.

**Figure 8 ijms-25-00509-f008:**
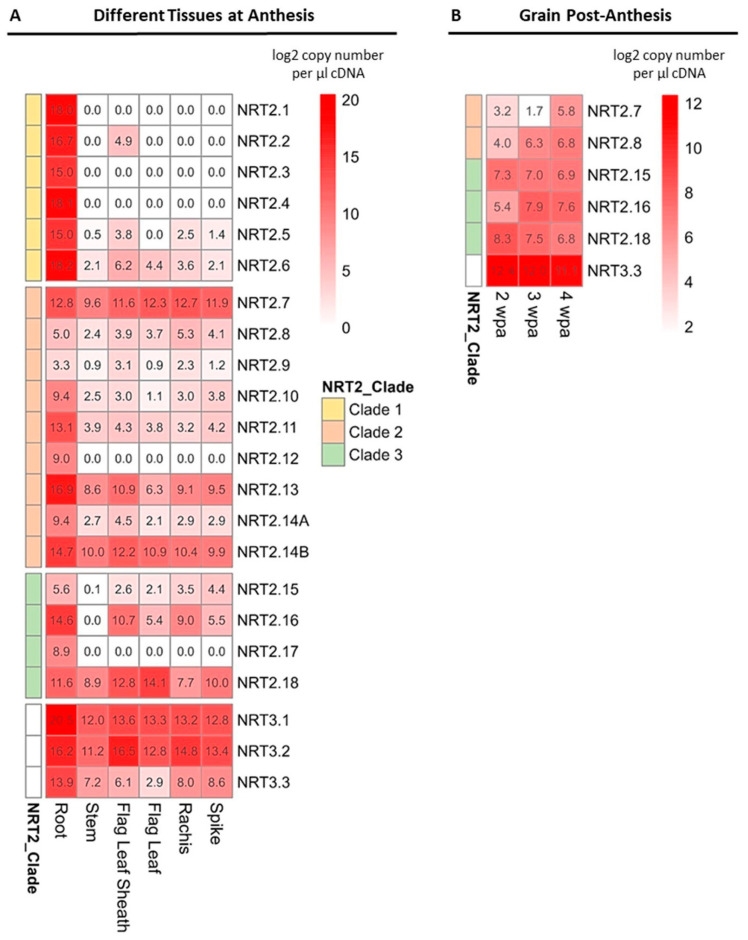
The expression patterns of the *NRT2* and *NRT3* genes in the specific tissues of field-grown wheat (cv Cadenza). The fertilizer application rate was 200 kg N ha^−1^. (**A**) Heatmap comparison of the *NRT2* and *NRT3* gene expressions in different tissues at anthesis and (**B**) in the grain at different stages post-anthesis. Expression analysis was performed by quantitative RT-qPCR, and the expression values are expressed as the log2 copy number per μL of cDNA. Data are the means of three biological replicates. wpa: weeks post-anthesis.

**Table 1 ijms-25-00509-t001:** Summary of the identified wheat *NRT2* and *NRT3* genes. This table provides details on the identified wheat *NRT2* and *NRT3* genes, including their accession numbers, chromosomal locations, and length of encoded proteins. The corresponding gene names from previous studies [26,27,28] were aligned with the suggested nomenclature.

Gene	GeneID	Chr	Start	End	Strand	Protein Length (aa)	[26]	[27]	[28]
*TaNRT2.1-6A*	TraesCS6A02G031200	6A	15,781,020	15,782,725	−1	509	*TaNRT2.1-A6*	*TaNRT2-6A6*	*TaNRT2-6A.6*
*TaNRT2.1-6B*	TraesCS6B02G044000	6B	26,591,111	26,592,640	1	509	*TaNRT2.1-B6*	*TaNRT2-6B6*	*TaNRT2-6B.1*
*TaNRT2.1-6D*	TraesCS6D02G035600	6D	14,618,629	14,620,585	1	509	*TaNRT2.1-D6*	*TaNRT2-6D6*	*TaNRT2-6D.1*
*TaNRT2.2-6A*	TraesCS6A02G031100	6A	15,765,759	15,767,783	1	507	*TaNRT2.1-A4*	*TaNRT2-6A5*	*TaNRT2-6A.5*
*TaNRT2.2-6B*	TraesCS6B02G044100	6B	26,596,252	26,597,775	−1	507	*TaNRT2.1-B4*	*TaNRT2-6B1*	*TaNRT2-6B.2*
*TaNRT2.2-6D*	TraesCS6D02G035800LC	6D	14,624,460	14,625,647	−1	395	*NA*	*NA*	*TaNRT2-6D.2*
*TaNRT2.3-6A*	TraesCS6A02G031000	6A	15,756,560	15,758,437	1	507	*TaNRT2.1-A3*	*TaNRT2-6A4*	*TaNRT2-6A.4*
*TaNRT2.3-6B*	TraesCS6B02G044200	6B	26,616,491	26,618,567	−1	507	*TaNRT2.1-B3*	*TaNRT2-6B4*	*TaNRT2-6B.3*
*TaNRT2.3-6D*	TraesCS6D02G035700	6D	14,631,385	14,633,069	−1	507	*TaNRT2.1-D3*	*TaNRT2-6D4*	*TaNRT2-6D.3*
*TaNRT2.4-6A*	TraesCS6A02G030900	6A	15,747,526	15,749,383	1	507	*TaNRT2.1-A5*	*TaNRT2-6A3*	*TaNRT2-6A.3*
*TaNRT2.4-6B*	TraesCS6B02G044300	6B	26,625,403	26,626,926	−1	507	*TaNRT2.1-B5*	*TaNRT2-6B3*	*TaNRT2-6B.4*
*TaNRT2.4-6D*	TraesCS6D02G035800	6D	14,655,066	14,656,589	−1	507	*TaNRT2.1-D5*	*TaNRT2-6D3*	*TaNRT2-6D.4*
*TaNRT2.5-6A*	TraesCS6A02G030800	6A	15,734,520	15,736,043	1	507	*TaNRT2.1-A2*	*TaNRT2-6A2*	*TaNRT2-6A.2*
*TaNRT2.5-6B*	TraesCS6B02G044400	6B	26,633,039	26,634,966	−1	565	*TaNRT2.1-B2*	*TaNRT2-6B2*	*TaNRT2-6B.5*
*TaNRT2.5-6D*	TraesCS6D02G036100LC	6D	14,662,011	14,664,928	−1	305	*NA*	*NA*	*TaNRT2-6D.5*
*TaNRT2.6-6A*	TraesCS6A02G030700	6A	15,727,844	15,729,367	1	507	*TaNRT2.1-A1*	*TaNRT2-6A1*	*TaNRT2-6A.1*
*TaNRT2.6-6B*	TraesCS6B02G044500	6B	26,644,113	26,645,632	−1	485	*TaNRT2.1-B1*	*TaNRT2-6B5*	*TaNRT2-6B.6*
*TaNRT2.6-6D*	TraesCS6D02G035900	6D	14,679,252	14,680,775	−1	507	*TaNRT2.1-D1*	*TaNRT2-6D2*	*TaNRT2-6D.6*
*TaNRT2.7-6A*	TraesCS6A02G032400	6A	15,951,566	15,953,536	−1	508	*TaNRT2.2-A2-1*	*TaNRT2-6A7*	*TaNRT2-6A.7*
*TaNRT2.7-6B*	TraesCS6B02G045600	6B	27,122,861	27,124,387	−1	508	*TaNRT2.2-B2-1*	*TaNRT2-6B7*	*TaNRT2-6B.7*
*TaNRT2.7-6D*	TraesCS6D02G037200	6D	15,383,513	15,385,158	−1	508	*TaNRT2.2-D2-1*	*TaNRT2-6D7*	*TaNRT2-6D.7*
*TaNRT2.8-6A*	TraesCS6A02G032500	6A	16,098,637	16,100,163	−1	508	*TaNRT2.2-A2-2*	*TaNRT2-6A8*	*TaNRT2-6A.8*
*TaNRT2.8-6B*	TraesCS6B02G045700	6B	27,169,710	27,171,230	−1	506	*TaNRT2.2-B2-2*	*TaNRT2-6B8*	*TaNRT2-6B.8*
*TaNRT2.8-6D*	TraesCS6D02G037300	6D	15,418,086	15,419,612	−1	508	*TaNRT2.2-D2-2*	*TaNRT2-6D8*	*TaNRT2-6D.8*
*TaNRT2.9-6B*		6B	27,180,298	27,181,751	−1		*NA*	*NA*	*NA*
*TaNRT2.9-6D*		6D	15,426,752	15,427,967	−1		*NA*	*NA*	*NA*
*TaNRT2.10-6A*	TraesCS6A02G032800	6A	16,357,746	16,359,603	1	507	*TaNRT2.2-A1*	*TaNRT2-6A9*	*TaNRT2-6A.9*
*TaNRT2.10-6B*	TraesCS6B02G046500	6B	27,685,182	27,687,046	1	507	*TaNRT2.2-B1*	*TaNRT2-6B9*	*TaNRT2-6B.9*
*TaNRT2.10-6D*	TraesCS6D02G037800	6D	15,658,356	15,659,879	1	507	*TaNRT2.2-D1*	*TaNRT2-6D9*	*TaNRT2-6D.9*
*TaNRT2.11-6A*	TraesCS6A02G032900	6A	16,374,353	16,376,212	−1	509	*TaNRT2.2-A4-1*	*TaNRT2-6A10*	*TaNRT2-6A.10*
*TaNRT2.11-6B*	TraesCS6B02G059100LC	6B	27,709,217	27,771,770	−1	197	*NA*	*NA*	*NA*
*TaNRT2.11-6D*	TraesCS6D02G037900	6D	15,696,840	15,698,711	−1	509	*TaNRT2.2-D4-1*	*TaNRT2-6D10*	*TaNRT2-6D.10*
*TaNRT2.11-U*	TraesCSU02G507900LC	U	356,326,164	356,329,757	1	197	*NA*	*NA*	*NA*
*TaNRT2.12-6D*	TraesCS6D02G038000	6D	15,710,039	15,711,562	−1	507	*TaNRT2.2-D4-2*	*TaNRT2-6D11y*	*TaNRT2-6D.11*
*TaNRT2.13-6A*	TraesCS6A02G033000	6A	16,386,427	16,388,254	−1	509	*TaNRT2.2-A4-2*	*TaNRT2-6A11*	*TaNRT2-6A.11*
*TaNRT2.13-6B*	TraesCS6B02G046600	6B	27,778,038	27,779,912	−1	509	*TaNRT2.2-B4*	*TaNRT2-6B11*	*TaNRT2-6B.10*
*TaNRT2.13-6D*	TraesCS6D02G038100	6D	15,745,837	15,748,091	−1	509	*TaNRT2.2-D4-3*	*TaNRT2-6D11x*	*TaNRT2-6D.12*
*TaNRT2.14A-6A*	TraesCS6A02G033100	6A	16,398,961	16,400,795	−1	508	*TaNRT2.2-A3-1*	*TaNRT2-6A12*	*TaNRT2-6A.12*
*TaNRT2.14A-6B*	TraesCS6B02G059400LC	6B	27,805,416	27,805,802	−1	128	*NA*	*NA*	*NA*
*TaNRT2.14A-6D*	TraesCS6D02G038200	6D	15,797,524	15,799,374	−1	508	*TaNRT2.2-D3-1*	*TaNRT2-6D13*	*TaNRT2-6D.13*
*TaNRT2.14B-6A*	TraesCS6A02G033200	6A	16,408,185	16,410,137	−1	508	*TaNRT2.2-A3-2*	*TaNRT2-6A13*	*TaNRT2-6A.13*
*TaNRT2.14B-6B*	TraesCS6B02G046700	6B	27,818,480	27,820,351	−1	508	*TaNRT2.2-B3*	*TaNRT2-6B13*	*TaNRT2-6B.11*
*TaNRT2.14B-6D*	TraesCS6D02G038300	6D	15,807,091	15,808,955	−1	508	*TaNRT2.2-D3-2*	*TaNRT2-6D12*	*TaNRT2-6D.14*
*TaNRT2.14C-6D*		6D	15,781,861	15,784,445	−1	501	*NA*	*NA*	*NA*
*TaNRT2.15-2A*	TraesCS2A02G074800	2A	33,054,150	33,056,031	1	502	*TaNRT2.3-A1*	*TaNRT2-2A1*	*TaNRT2-2A*
*TaNRT2.15-2D*	TraesCS2D02G073500	2D	30,787,486	30,789,242	1	499	*TaNRT2.3-D1*	*TaNRT2-2D1*	*TaNRT2-2D*
*TaNRT2.16-3A*	TraesCS3A02G254000	3A	475,304,797	475,306,341	−1	514	*TaNRT2.4-A1*	*TaNRT2-3A1*	*TaNRT2-3A*
*TaNRT2.16-3B*	TraesCS3B02G285900	3B	457,633,984	457,635,782	−1	514	*TaNRT2.4-B1*	*TaNRT2-3B1*	*TaNRT2-3B*
*TaNRT2.16-3D*	TraesCS3D02G254900	3D	356,623,041	356,624,585	−1	514	*TaNRT2.4-D1*	*TaNRT2-3D1*	*TaNRT2-3D*
*TaNRT2.17-U*	TraesCSU02G002800	(1B)	2,667,931	2,669,478	−1	515	*TaNRT2.4-3*	*TaNRT2-Un1*	*TaNRT2-U.1*
*TaNRT2.17-1D*	TraesCS1D02G035700	1D	16,504,613	16,506,169	1	518	*TaNRT2.4-2*	*TaNRT2-1D1*	*TaNRT2-1D*
*TaNRT2.18-7A*	TraesCS7A02G428500	7A	621,910,950	621,913,739	1	468	*TaNRT2.5-A1*	*TaNRT2-7A1*	*TaNRT2-7A*
*TaNRT2.18-7B*	TraesCS7B02G328700	7B	583,923,053	583,926,829	1	486	*TaNRT2.5-B1*	*TaNRT2-7B1*	*TaNRT2-7B*
*TaNRT2.18-7D*	TraesCS7D02G420900	7D	540,617,018	540,627,808	1	483	*TaNRT2.5-D1*	*TaNRT2-7D1*	*TaNRT2-7D*
*TaNRT3.1-6A*	TraesCS6A02G209900	6A	380,361,041	380,362,225	−1	198	*TaNAR2.1-A1*	*NA*	*NA*
*TaNRT3.1-6B*	TraesCS6B02G238700	6B	415,788,848	415,790,024	−1	198	*TaNAR2.1-B1*	*NA*	*NA*
*TaNRT3.1-6D*	TraesCS6D02G193100	6D	267,236,634	267,237,837	−1	198	*TaNAR2.1-D1*	*NA*	*NA*
*TaNRT3.2-6A*	TraesCS6A02G210000	6A	381,025,658	381,028,549	−1	251	*TaNAR2.2-A1*	*NA*	*NA*
*TaNRT3.2-6B*	TraesCS6B02G238800	6B	415,863,369	415,864,548	−1	198	*TaNAR2.2-B1*	*NA*	*NA*
*TaNRT3.2-6D*	TraesCS6D02G193200	6D	267,514,716	267,515,418	−1	198	*TaNAR2.2-D1*	*NA*	*NA*
*TaNRT3.3-4A*	TraesCS4A02G367300	4A	640,232,228	640,233,158	1	255	*TaNAR2.3-A1*	*NA*	*NA*
*TaNRT3.3-5B*	TraesCS5B02G719500LC	5B	671,187,506	671,188,471	−1	236	*NA*	*NA*	*NA*
*TaNRT3.3-5D*	TraesCS5D02G506100	5D	531,971,003	531,972,109	−1	199	*TaNAR2.3-D1*	*NA*	*NA*
*TaNRT3.4-5B*	TraesMAC5B03G03000180	5B	671,130,989	671,138,061	−1	181	*NA*	*NA*	*NA*
*TaNRT3.4-5D*	TraesMAC5D03G03214570	5D	531,900,482	531,902,366	−1	190	*NA*	*NA*	*NA*

## Data Availability

All data supporting the findings of this study are available within the paper and within its Appendix A, which are published online.

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
