# Peer review of "The Functional Diversity of the High-Affinity Nitrate Transporter Gene Family in Hexaploid Wheat: Insights from Distinct Expression Profiles"

_ijms, 2023, doi:10.3390/ijms25010509_

Round 1
Reviewer 1 Report
Comments and Suggestions for Authors
The present study focused on wheat NRT2 and NRT3 gene family. The authors identified 53 NRT2 and 11 NRT3 in wheat genome. Furthermore, they performed their expression analysis in detail. The results in this study provided insights into the potential roles of the NRT2s in the mechanisms uderlyin N uptake. It is very miportan for wheat breeding.
The objectives of this study are clear, the experimental design is well thought out, and the results are interesting. However, the introduction section is somewhat lacking.
1. Please introduce and explain the NRT2 and NTR3gene family, not only the findings in Arabidopsis thaliana, the model plant, but also the findings in other crops and the current status of research.
2. I would like to see the background of this research, e.g., the agronomic implications of this research for improving the yield potential or quality of wheat.
Furthermore,
3. Agronomic implications should also be added in the conclusion.
Reviewer 2 Report
Comments and Suggestions for Authors
The authors describe the genomic distribution and functionality of high-affinity nitrate transporters (NRT2 and NTR3) gene families in hexaploid wheat genome. These are key components of nitrogen acquisition and distribution within the plant. The manuscript includes three main parts: (1) bioinformatics and phylogenetic analysis of NRT genes in the wheat genome; (2) gene expression analysis in response to N stress, or to nitrate supply after N stress at the seedling stage; (3) gene expression patterns measured in the field under high N, measured at anthesis and maturation.
I recommend accepting the manuscript for publication with revision and addressing the following comments/questions:
1. Part 1 includes an excellent description of bioinformatics and phylogenetics analysis, and part 2 is also well described, both in the results + discussion. However, part 3 evaluates NRT gene expression in plants tested under a different experimental setup. The authors used T. aestivum cv Paragon, at the seedling stage tested by hydroponics while in part 3 they used Cadenza under a high N of 200kg in the field, in which plants are sampled at anthesis. The result of this part is hardly mentioned in the discussion part, and it seems unconnected with the previous part.
2. Up-regulation of NRT genes can explain mechanisms associated with N acquisition or utilization but it may not necessarily improve NUE, which is a complex trait and calculated based on grain yield. Furthermore, there is no phenotyping or comparison between high and low NUE genotypes, that can relate between gene expression and NUE. The authors should clarify the association between NRT and NUE.
Reviewer 3 Report
Comments and Suggestions for Authors
N is one of the main necessary elements for the normal development of a plant. Problems associated with its absorption, transport and localization are of important practical significance for agriculture. The authors of the presented manuscript did a great deal of work studying the NRT families in wheat, identified all members of the NRT2 and NRT3 families from wheat, compared them with members of these families from other plant species and made their classification. The authors confirmed their hypothesis about the involvement of NRT family genes in either N transport or N homeostasis in wheat grown under field conditions. This increases the value of the results obtained.
There are some minor comments regarding the design of the manuscript.
233- it is more correct to use cDNA rather than mRNA, since the concentration was determined precisely by cDNA
Figures 2 and 3. - what does the value 0.1 mean?
Do Ffisher's LSD values refer to what incubation time or are these values constant?
315 - space between 10 mM
For some reason, the same reference is given in different ways. Prepare according to journal requirements
The temperature in the Materials and Methods section is the same.
649 -NO3-
864 - remove space
Reviewer 4 Report
Comments and Suggestions for Authors
Sigalas et al conducted a study to understand the responses of NRT2 and NRT3 gene families to nitrogen availability. The manuscript is well presented and provides information. However, there are few crucial points that if considered will increase the value of the manuscript and may be readability.
-First of all, please mention some part of methodology in the abstract. Please summarize what has been done to achieve the mentioned aim.
-Please provide information on the quality of isolated RNAs. For example, their gel picture or RIN values. You can cite proper studies while explaining the quality of the isolated RNA. For example, https://www.mdpi.com/2073-4395/13/3/631 or https://www.mdpi.com/2073-4395/12/10/2421. RNA quality is important for such studies. Any contamination may affect the reliability of the obtained results.
-How many replicates were used for RNA isolation and LECO analysis? Please mention about the statistical analysis conducted for the nitrogen analysis.
- In the caption of Figure 8, please add the situation of nitrogen availability.
I do believe that the manuscript can be accepted once the authors address the mentioned points and enrich the manuscript with the crucial information.
Comments on the Quality of English LanguageMinor editing of English language required
Round 2
Reviewer 4 Report
Comments and Suggestions for Authors
The manuscript can be accepted in its present form.
Comments on the Quality of English LanguageMinor editing of English language required